**Technical note: Towards a stronger observational support for haze pollution control by interpreting carbonaceous aerosol results derived from different measurement approaches**

Yuan Cheng, Ying-jie Zhong, Zhi-qing Zhang, Xu-bing Cao, Jiu-meng Liu[*]

State Key Laboratory of Urban Water Resource and Environment, School of Environment, Harbin Institute of Technology, Harbin, 150090, China

[*] Corresponding author. Jiu-meng Liu (jiumengliu@hit.edu.cn).

**Abstract**

As China's fine particulate matter ($PM_{2.5}$) has decreased nationwide during the last decade, further improvement of air quality became more challenging, imposing higher requirements on the observational support for the understanding of aerosol sources. This was particularly the case for the severe cold climate region in Northeast China, which suffered from relatively slow decreasing rate and high exposure risk of $PM_{2.5}$. Here we evaluated carbonaceous aerosol data measured by different sampling and analytical approaches, based on field campaigns conducted during a frigid winter and an agricultural-fire impacted spring in Harbin. For both the high- and low-volume sampling, a total of four sets of organic and elemental carbon results were derived by applying two commonly-used temperature protocols (IMPROVE-A, i.e., IMPV, and NIOSH) to both untreated filters and those extracted by methanol. Only the IMPV-based results measured before the extraction were found to be indicative of aerosol sources, e.g., in reasonable accordance with secondary aerosol formation in winter and open burning impacts in spring. Thus the analytical method of IMPROVE-A on untreated samples was recommended for future field observations and source apportionments of $PM_{2.5}$ in the studied region. In addition, although the low- and high-volume samplers typically led to comparable measurement results for various species, exceptions were identified for water-soluble potassium ($K^+$) and some fire-emitted chromophores. We suggested that K+ and light

24    absorption coefficients of brown carbon should be compared or integrated with caution across

studies using different PM$_{2.5}$ samplers.

## 1. Introduction

Carbonaceous aerosols are a complex mixture of compounds exhibiting a gradual change in chemical and physical properties (Pöschl, 2005; Andreae and Gelencsér, 2006), e.g., from colorless organics with low molecular weights, to "dark" brown carbon with relatively high thermal stabilities (Chakrabarty et al., 2023), and finally to refractory black carbon which is strongly light-absorbing. As important contributors to both fine particulate matter ($PM_{2.5}$) pollution and radiative forcing, they have long been targeted to achieve a synergistic improvement of air quality and mitigation of climate change (Fuzzi et al., 2015; von Schneidemesser et al., 2015; Liu et al., 2022). However, it remains a challenge to properly represent carbonaceous aerosols in chemical transport models, as each step along the way between the estimations of sources and tempo-spatial variations is difficult. For example, considerable uncertainties exist in the open burning emission, secondary organic aerosol (SOA) budget and black carbon lifetime (Andela et al., 2022; Chang et al., 2022; Zhong et al., 2023). This in turn introduces substantial uncertainties to the climate and health effects of carbonaceous aerosols (Li et al., 2022).

Field observational data on carbonaceous aerosols, including those derived from ground and aircraft measurements, are essential to constrain the simulation results and subsequently to improve the model performance (Philip et al., 2014; Wang et al., 2014b, 2018; Gao et al., 2022; Eckhardt et al., 2023). Relying on on-line instruments such as the Single Particle Soot Photometer (SP2), aircraft studies typically covered relatively short periods (e.g., up to about one month) but provided measurement results with high time and spatial resolutions (Samset et al., 2014). Offline and semi-continuous techniques (e.g., lab and field carbon analyzers for elemental carbon) were more commonly used in ground observations, producing datasets with relatively low time resolutions but

spanning several months to decades (Dao et al., 2019; Hand et al., 2024). After accounting for the
difference in time resolution, the integration of carbonaceous aerosol data across studies and regions
was still not straightforward. A major obstacle was caused by the multitude of measurement
principles (Petzold et al., 2013), which was intensively reflected by the considerable and, more
importantly, variable discrepancies in black carbon results among different methods (Buffaloe et al.,
2014; Li et al., 2019; Pileci et al., 2021; Tinorua et al., 2024). This problem was far from being
properly addressed, although great efforts have been devoted to refine the respective measurement
approach such as the thermal-optical (Cavalli et al., 2010), optical (Collaud Coen et al., 2010) and
SP2 (Laborde et al., 2012) techniques. In addition, this problem was to some extent overlooked in
China, which might be partially responsible for the inconsistent source apportionment results
obtained by different studies. For example, both Zheng et al. (2015) and Liu et al. (2020) applied
the elemental carbon (EC) tracer method to estimate secondary OC (SOC) during winter in Beijing,
but the two studies derived conflicted conclusions on the contribution of heterogeneous chemistry
to SOC formation (i.e., minimal vs. significant) since different analytical methods (i.e., NIOSH vs.
IMPROVE-A temperature protocols) for EC were deployed. Such inconsistencies substantially
weakened the observational support for the understanding of aerosol sources and thus the control of
haze pollution.
With a considerable decrease in the national $PM_{2.5}$ since 2013, it became more challenging to
further improve the air quality in China (Cheng et al., 2021). This imposed higher requirements on
the observational insights into aerosol sources, including the evaluation of carbonaceous aerosol
results among various measurement approaches. Here we focused on the widely-used thermal-
optical method, which separates carbonaceous components into two fractions with different thermal
stabilities and light absorption capacities, i.e., organic carbon (OC) and EC. The basis of the
separation includes two points: EC evolves form the filter at higher temperatures than OC, and the
removal of EC leads to a rapid increase in the filter transmittance and reflectance signals. A major
problem in this method is that a considerable fraction of OC could be transformed into char-OC,
which is difficult to be robustly distinguished from EC with respect to either thermal or optical
behavior. In addition, the amount and optical properties of char-OC were found to depend on the
temperature protocol deployed (Yu et al., 2002; Yang and Yu, 2002; Subramanian et al., 2006). This
to a large extent explained the EC discrepancies among various protocols. However, it remained
unclear how the charring process and thus the EC measurement were influenced by OC sources and
composition (Chiappini et al., 2014). In addition, to reduce or minimize the interference from char-
OC, several investigators have tried to remove a fraction of OC from the samples before thermal-
optical analysis, by extracting the filters using water, methanol or other solvents (Piazzalunga et al.,
2011; Giannoni et al., 2016; Lappi and Ristimäki, 2017; Aakko-Saksa et al., 2018; Hu et al., 2023).
However, inconsistent patterns were identified for the effects of OC removal on EC determination,
with evidences available for both an increase (e.g., Piazzalunga et al., 2011) and a decrease in EC
(e.g., Hu et al., 2023) after the extraction. The discussions above indicated that the thermal-optical
methods, including the practicability of sample pretreatment by extraction, merit further
investigations.

In this study, we compared carbonaceous aerosol results determined by different analytical as

well as sampling approaches, based on filter samples collected in Harbin, the northernmost megacity
in China. Compared to other megacities such as Beijing, Harbin is characterized by the frigid winter
(with an average temperature of about –20 °C in January) and the massive agricultural sector in
surrounding areas (i.e., the Songnen Plain in Northeast China). In addition, Harbin and other cities
in Northeast China have largely been overlooked in clean air actions and thus studies on haze, as
indicated by the limited observational data available (Liu et al., 2022). This lack of investigation
was partially responsible for the relatively slow decreasing rate (Xiao et al., 2022) and high exposure
risk (Wei et al., 2023) of $PM_{2.5}$ in this distinct region. Thus our study on measurement methods of
carbonaceous aerosols is expected to be a support for future efforts on the exploration of $PM_{2.5}$
sources in Northeast China, which are essential for further improving the regional air quality.
**2. Methods**
**2.1 Field sampling**
$PM_{2.5}$ samples were collected at an urban site in Harbin, i.e., on the campus of Harbin Institute
of Technology, during the winter and spring of 2021. The sampling was done by a mass flow
controlled high-volume sampler (TE-6070BLX-2.5-HVS; Tisch Environmental, Inc., OH, USA)
and a low-volume sampler (MiniVol; Airmetrics, OR, USA), operated using quartz-fiber filters (Pall
Corporation, NY, USA) at flow rates of 1.13 $m^3$/min and 5 L/min, respectively. The flow rates,
together with the particle-laden filter areas, could be translated into the face velocities of 46.34 and
7.35 cm/s for the high- and low-volume (HV and LV) samplers, respectively. This indicated that
when the two samplers were run in parallel, the HV-to-LV ratio of particle loading would be 6.3.
The 2021 winter campaign covered the entirety of January, the coldest month during that year
with an average temperature of −19 °C. In addition, the spring campaign was conducted during 10–
30 April of 2021, a period with frequent occurrences of agricultural fires (as indicated by the
satellite-based active fire detection results; Figure S1). For both seasons, the HV sampler was used
to collect daytime (09:00–16:00) and nighttime (21:00–05:00 of the next day) samples, while the
LV one was operated on a daily basis (~09:00–09:00 of the next day), leading to 24-h integrated
samples. Each LV sample generally corresponded to two HV samples, although the two samplers
were not exactly parallel. One reason for the relatively short sampling durations of HV was to avoid
high particle loadings that could prohibit proper filter transmittance measurement (Lappi and
Ristimäki, 2017).
**2.2 Laboratory analysis**
For both the HV and LV samples, two punches were prepared to determine OC and EC using
a thermal-optical carbon analyzer (DRI-2001; Atmoslytic Inc., CA, USA). One punch was directly
measured, while the other one was immersed in methanol (HPLC grade; Fisher Scientific Company
L.L.C., NJ, USA) for an hour without stirring or sonication, dried in air for another hour, and then
analyzed. All the pairs of untreated and extracted punches were measured using the IMPROVE-A
and NIOSH temperature protocols, both of which were operated with transmittance charring
correction (Figure 1). This correction approach was applied since the intensity of the filter
transmittance signal ($I$) has a clear association with EC, e.g., as assumed by the Aethalometer,
another widely used instrument for measuring black carbon. Inter-protocol comparisons showed
good repeatability for both the total carbon (TC) and optical attenuation (ATN) results (Figure 2),
demonstrating the robustness of the analyzer for detecting carbon and filter transmittance signals.
Here ATN was calculated as $\ln(I_{final}/I_{initial})$, where $I_{initial}$ and $I_{final}$ indicate $I$ measured at beginning
(i.e., when the particle-laden filter has not been heated) and end (i.e., when all the deposited carbon
has been combusted off the filter) of thermal-optical analysis, respectively. ATN was of interest
because it was closely related to EC loading ($EC_s$, in $\mu g/cm^2$), e.g., typically with a linear
dependence for relatively low $EC_s$ levels (Chen et al., 2020; Liu et al., 2020). It should be noted that
for the parallel TC and ATN measurements by different protocols, the relative standard deviation
(RSD) levels indeed increased after the extraction, e.g., from ~2 to 5% and from ~2 to 4% for the
HV samples, respectively. However, the RSD levels, i.e., the uncertainties, were considered low
enough for both the untreated and extracted filters.

In addition, following the method developed by Hecobian et al. (2010), wavelength-resolved

light absorption coefficients ($b_{abs}$) of the methanol extracts, i.e., the dissolved brown carbon (BrC),
were measured using a spectrophotometer (Ocean Optics Inc., FL, USA) coupled with a 2.5-m long
liquid waveguide capillary cell (LWCC; World Precision Instruments, FL, USA). Inorganic ions and
levoglucosan were also determined for the HV and HV samples, by analyzing their water extracts
using a Dionex ion chromatography system (ICS-5000$^+$; Thermo Fisher Scientific Inc., MA, USA).

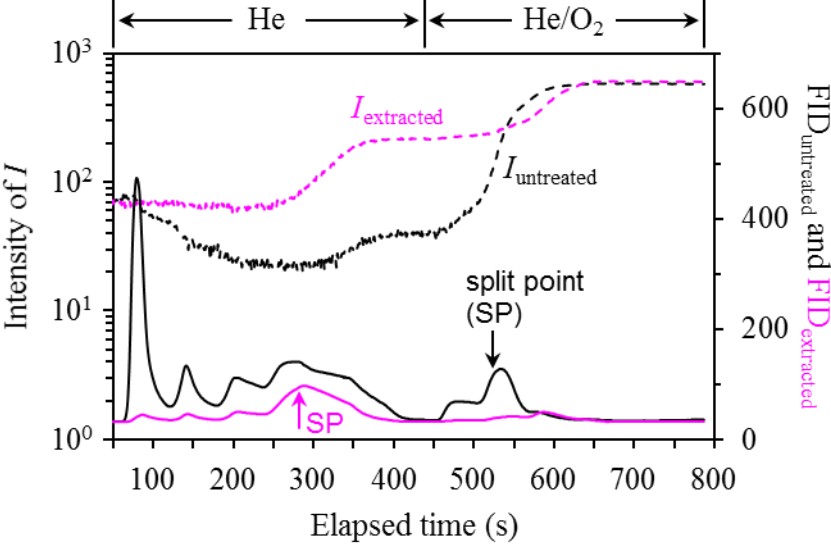


**Figure 1.** Thermograms measured using a pair of untreated and extracted HV filters. The sample
was collected during the daytime of 25 January, 2021. Temperature protocol used was NIOSH, in
which the filter was heated in a He (first to 870 ℃ stepwise and then cooled down to 550 ℃) and a
He/O$_2$ (from 550 to 890 ℃ stepwise) atmosphere sequentially. NIOSH had fixed durations for the
various heating stages and thus was preferred for the comparison of thermograms. $I$ indicates the
filter transmittance signal; FID indicates the carbon signal, which was measured by a flame
ionization detector. The subscripts "untreated" and "extracted" distinguished the thermograms
measured before and after the extraction, while the split points of OC and EC were marked by the
arrows.

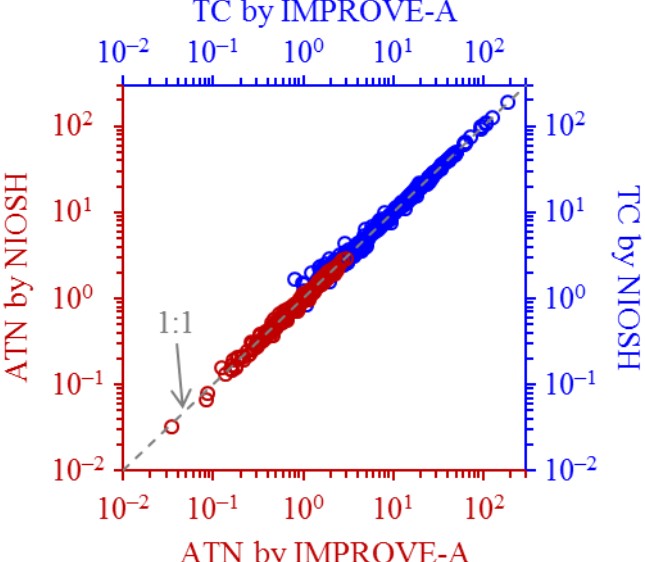


**Figure 2.** Comparisons of TC (in μgC/cm$^2$) and ATN (dimensionless) measured using different temperature protocols. Results from the HV and LV samples, both untreated and extracted, were combined for the comparisons. Linear regression of the NIOSH-based TC on that determined by IMPROVE-A led to a slope of 0.99 ±0.00 ($r$ = 1.00; intercept was set as zero). Similar regression results (i.e., slope = 0.99 ±0.00; $r$ = 1.00) were obtained for ATN. The good repeatabilities on one hand demonstrated the performance of carbon analyzer for measuring the carbon and laser transmittance signals, and on other hand indicated a homogeneous distribution of carbonaceous components for not only the untreated but also the extracted filters, i.e., a negligible disturbance of EC as well as other insoluble carbon by the extraction.

**2.3 Open-access data**

Meteorological data (e.g., temperature and relative humidity) and air quality data (e.g., PM$_{2.5}$, PM$_{10}$ and CO) for the measurement periods were obtained with a time resolution of 1 h from Weather Underground (https://www.wunderground.com/) and the China National Environmental Monitoring Center (CNEMC; https://air.cnemc.cn:18007/), respectively.

**3. Results and discussion**

**3.1 Evaluation of EC results from the winter campaign**

A precondition for proper separation of OC and EC is that the filter transmittance signal ($I$) could properly reflect the formation of light-absorbing char-OC during the inert mode (which would result in a decrease in $I$), and the combustion, i.e., removal, of char-OC and EC during the oxidation

mode (which would result in an increase in *I*). An empirical approach to evaluate this precondition
is to examine the dependence of ATN on $EC_s$ (Subramanian et al., 2006). A linear relationship was
typically observed for relatively low $EC_s$ levels and in this case, the precondition was commonly
believed to be valid. However, the linearity did not necessarily extend when $EC_s$ further increased,
since previous studies frequently found that the measured ATN could be considerably lower than
expected for heavily-loaded samples (Shen et al., 2013; Costa et al., 2016; Chen et al., 2020). The
deviation of ATN vs. $EC_s$ dependence from linear relationship was usually termed the loading effect,
and a traditional explanation was that ATN became less sensitive to the variation of $EC_s$ as filter
loading increased. An extreme case was observed during winter in Beijing, that the ATN values
were largely unchanged for heavily-loaded filters with TC varying between 150 and 300 $\mu gC/cm^2$
(Liu et al., 2019). For the samples showing non-linear ATN vs. $EC_s$ dependence, their EC results
should be interpreted with caution.

We first investigated the relationship between ATN and $EC_s$ for the wintertime HV samples,

focusing on the results from IMPROVE-A. For the untreated filters, ATN correlated linearly with
$EC_s$ (leading to a regression slope of 42.8 $\pm$ 1.9 $m^2/gC$ and a close-to-zero intercept; $r = 0.95$) when
the filters were lightly to moderately loaded, i.e., when the $EC_s$ levels were below 5 $\mu gC/cm^2$ (Figure
3a). The physical meaning of the slope was the mass absorption efficiency (MAE) of black carbon,
but with artifacts such as that caused by the multiple scattering effect (Lack et al., 2014). The overall
impact of various artifacts results in an overestimation of MAE, typically by factors of ~3 (Knox et
al., 2009; Qin et al., 2018). The linearity determined for the $EC_s$ range of below 5 $\mu gC/cm^2$ did not
hold for the more heavily loaded samples ($N = 3$, as highlighted by the solid circles in Figure 3a),
showing evidence for the loading effect. For the extracted samples, a linear correlation between
ATN and $EC_s$ was also identified for relatively low EC loadings (Figure 3b), with a similar
relationship (i.e., a regression slope of 41.5 $m^2/gC$ and a close-to-zero intercept; $r = 0.95$) to that
derived from the untreated filters. However, it was noteworthy that $EC_{max}$, the upper limit of EC
loading for a linear ATN vs. $EC_s$ dependence, was only 3 $\mu gC/cm^2$ for the extracted filters, much
lower than that determined for the untreated ones. Due to the shift of $EC_{max}$, 51% of the extracted
samples showed evidence for the loading effect, whereas this fraction was only 5% before extraction.

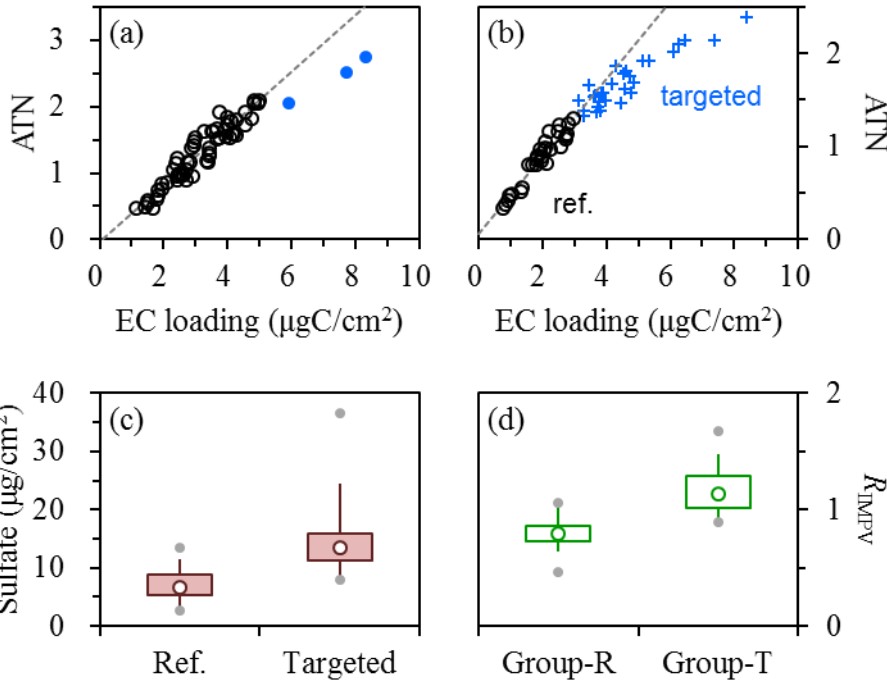


**Figure 3.** Relationships between ATN and EC loading, i.e., $EC_s$, for the **(a)** untreated and **(b)**
extracted HV filters collected in winter, using the IMPROVE-A protocol. Linear dependences were
observed for the untreated samples with $EC_s$ below 5 $\mu gC/cm^2$ and for the extracted samples with
$EC_s$ below 3 $gC/cm^2$, as indicated by the dashed lines in (a) and (b), respectively. Three untreated
samples had $EC_s$ above 5 $\mu gC/cm^2$, as highlighted by the solid circles in (a). In (b), the extracted
filters showing non-linear ATN vs. $EC_s$ dependence were termed the targeted samples;
correspondingly, the others were referred to as the reference ones (labelled as ref.). **(c)** Comparison
of sulfate loadings between the reference and targeted samples. **(d)** Comparison of the $EC_{extracted}$ to
$EC_{untreated}$ ratios, i.e., $R_{IMPV}$, between the reference and targeted groups of wintertime HV samples
(labelled as Group-R and Group-T, respectively). The targeted group indicated the targeted filters
and the corresponding untreated ones, while the reference group indicated the remaining pairs.
Lower and upper box bounds indicate the 25th and 75th percentiles, the whiskers below and above
the box indicate the 5th and 95th percentiles, the solid circles below and above the box indicate the
minimum and maximum, and the open circle within the box marks the median. All the EC results

involved were measured by IMPROVE-A.
The discussions above raised a question that why the extraction significantly reduced $EC_{max}$
for the HV samples. In principle, two factors could be responsible for the non-linear dependence of
ATN on $EC_s$, including gradual saturation of ATN with increasing filter loading (the traditional
explanation; Subramanian et al., 2006) and overestimation of EC mass. The $EC_{max}$ of untreated
samples corresponded to an ATN of 2.1, indicating that the saturation was presumably not a concern
for the ATN results below this value. Regarding the extracted samples showing evidence for the
loading effect (i.e., the targeted samples), ATN stayed below 2.1 for nearly all of them (28 out of
31), thus their non-linear ATN vs. $EC_s$ dependences should be primarily attributed to the
overestimation of EC mass rather than the saturation of ATN. Compared to the other extracted
samples, the targeted ones were characterized by substantially higher sulfate loadings (Figure 3c).
It was inferred that in addition to EC, the abundant sulfate was also a non-negligible contributor to
ATN (e.g., through backward scattering; Petzold et al., 2005; Collaud Coen et al., 2010). Thus when
the targeted samples were heated in the carbon analyzer, volatilization of sulfate would lead to a
decrease in ATN, i.e., an increase in filter transmittance signal. This was expected to result in a
premature split of OC and EC, and eventually an overestimation of EC. Other scattering components
such as nitrate and secondary organic aerosol (SOA) were not discussed here, since they were
typically considered soluble in methanol and should be absent in the extracted filters.
Comparison of EC between the targeted samples and the corresponding untreated ones (i.e.,
the targeted group) showed an overall increasing trend after the extraction (Figure 3d). For these
pairs of wintertime HV filters, the ratio of $EC_{extracted}$ (i.e., EC measured in the extracted samples) to
$EC_{untreated}$ (i.e., EC measured in the untreated samples) averaged 1.16 ±0.20. The extraction-induced
increase in EC coincided with the overestimation of elemental carbon mass by $EC_{extracted}$, which was
inferred to be associated with the presence of abundant sulfate in the extracted filters.
For the other pairs of wintertime HV samples (i.e., the reference group), the $EC_{extracted}$ to
$EC_{untreated}$ ratios averaged 0.80 $\pm$0.12, pointing to a decrease in EC after the extraction (Figure 3d).
This was also the case for all the LV samples collected during the winter campaign, with comparable
$EC_{extracted}$ to $EC_{untreated}$ ratios (averaging 0.78 $\pm$0.12). Here the LV samples were not divided into
subgroups because non-linear dependence of ATN on $EC_s$ was identified neither before nor after the
extraction (Figure S2). Given that the loss of insoluble carbon (e.g., EC) was negligible for our
extraction procedures (Figure 2 and Cheng et al., 2024), the extraction-induced decrease of EC
likely pointed to the underestimation of elemental carbon mass by $EC_{extracted}$. A common feature for
the HV samples in the reference group and the entirety of the LV samples was the relatively low
sulfate loadings. Cheng et al. (2024) inferred that small amounts of sulfate likely favored the
transmission of light through the extracted filters (e.g., by forward scattering; Petzold et al., 2005;
Collaud Coen et al., 2010). In this case, when the extracted samples were heated during thermal-
optical analysis, volatilization of sulfate would induce a drop of filter transmittance signal, which
could not be distinguished from that caused by the formation of char-OC. This was expected to
result in an overcorrection for char-OC, i.e., an underestimation of EC.
The contrasting $EC_{extracted}$ to $EC_{untreated}$ ratios observed for the two groups of wintertime HV
samples suggested that the influence of sulfate on the transmittance signal of the extracted filter was
likely loading-dependent. The influence was inferred to be dominated by backward scattering with
relatively high sulfate loadings (e.g., for the targeted group), whereas by forward scattering when
sulfate was less abundant (e.g., for the reference group). This inference was supported by the

comparison of evolution patterns of filter transmittance signal under different sulfate loadings

(Figure 4). For the extracted filter with abundant sulfate (i.e., Sample-A in Figure 4), the

transmittance signal was largely unchanged during the He mode despite the sufficient organic

carbon loading. Correspondingly, the operationally-defined char-OC only accounted for a relatively

small fraction of the carbon evolving during the $He/O_2$ mode (i.e., $He/O_2$ carbon). A possible

explanation was that as the sample was heated, the drop of $I$ induced by char-OC was compensated

by the increase of $I$ due to the reduction in sulfate-driven backward scattering. For the extracted

filter with relatively small amount of sulfate (i.e., Sample-B in Figure 4), however, the transmittance

signal decreased significantly during the He mode, and the char-OC contribution to $He/O_2$ carbon

became more considerable correspondingly. Given the much lower organic carbon loading for this

sample (e.g., ~70% lower than Sample-A), the decrease of $I$ was likely contributed by not only the

formation of char-OC but also the reduction in sulfate-driven forward scattering. The $EC_{extracted}$

results appeared to be biased by different artifacts in the high- and low-sulfate cases, resulting in

overestimations or underestimations of elemental carbon mass, respectively. The sulfate-induced

artifacts for $EC_{extracted}$ could be more directly reflected by the positive dependence of the $EC_{extracted}$

to $EC_{untreated}$ ratio on sulfate loading. As shown in Figure 5, the turning point for the artifact shifting

from an underestimation to overestimation of elemental carbon mass by $EC_{extracted}$ occurred in the

sulfate loading range of 10–15 $\mu g/cm^2$. Figure 5 also suggested that the artifacts for $EC_{extracted}$ were

difficult to be accounted for, e.g., by a constant correction factor. This prohibited the use of $EC_{extracted}$

for further analysis of aerosol composition and sources.

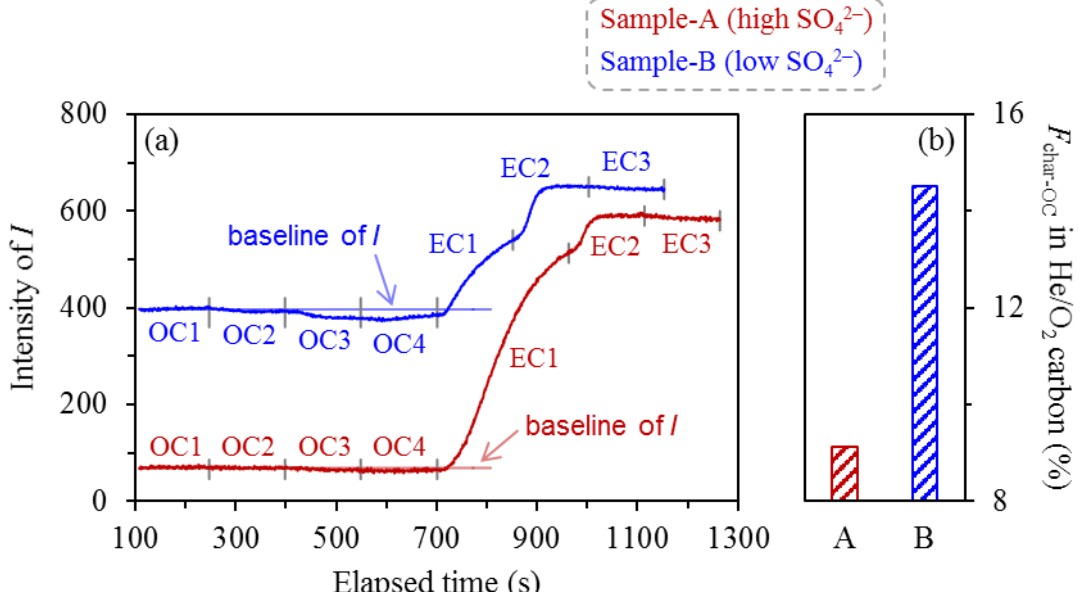

**Figure 4.** Comparisons of **(a)** the evolution patterns of filter transmittance signal ($I$) and **(b)** the fractions of char-OC (i.e., $F_{char\text{-}OC}$) in He/$O_2$ carbon for two extracted HV samples with relatively high and low sulfate loadings (namely Sample-A and Sample-B, respectively). The two samples were collected during the daytime of 25 January and the nighttime of 6 January, 2021, respectively. They had sulfate loadings of 13.21 and 3.29 μg/cm², and organic carbon loadings of 3.10 and 0.86 μgC/cm², respectively. The temperature protocol used was IMPROVE-A, in which the filter was first heated to 580 ℃ in a He atmosphere and then to 840 ℃ in a He/$O_2$ atmosphere. The two modes had 4 (i.e., OC1 to OC4) and 3 (i.e., EC1 to EC3) heating stages, respectively. He/$O_2$ carbon indicated the amount of carbon evolving during the oxidizing mode, and was typically comprised of char-OC and EC for IMPROVE-A.

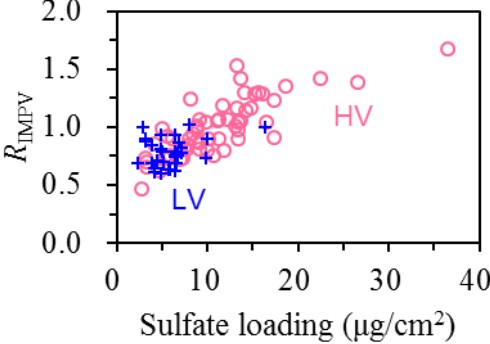

**Figure 5.** Dependence of $R_{IMPV}$, i.e., the $EC_{extracted}$ to $EC_{untreated}$ ratio determined by IMPROVE-A, on sulfate loading in winter. Consistent trends were observed by the HV and LV samples. The relatively wide range of $R_{IMPV}$ (approximately 0.5–1.5) provided solid evidence for the invalidation of the $EC_{extracted}$ to $EC_{untreated}$ ratio as an indicator for the extraction-induced loss of EC.

For the untreated filters, ATN exhibited a strong linear correlation with $EC_s$ for both HV and

LV, with three heavily-loaded HV samples as the only exception. Actually, for the three samples,

their ATN vs. $EC_s$ relationships did not deviate markedly from the regression line determined for
the lower $EC_s$ loadings. Therefore, we suggested that when applying IMPROVE-A to the winter
samples, the measurement uncertainties should be less significant for $EC_{untreated}$ compared to
$EC_{extracted}$.
The same conclusion could be reached by interpreting the OC to EC ratios (OC/EC). It has
been widely accepted that OC/EC depended strongly on SOA formation, after excluding the events
impacted by irregular emissions such as fireworks and open burning. Such events were not evident
throughout the winter campaign, and thus OC/EC was expected to increase with the enhancement
of secondary aerosol production. Here we used the relative abundance of secondary inorganic ions
(sulfate, nitrate and ammonium, i.e., SNA) compared to carbon monoxide (a typical primary
species), i.e., the SNA/CO ratio, as an indicator for the significance of secondary aerosols. A benefit
of using SNA/CO was that it was independent of EC measurement. The OC/EC ratio corresponding
to $EC_{untreated}$ [i.e., $(OC/EC)^*$] was determined directly by the thermal-optical results from the
untreated samples. For $EC_{extracted}$, the corresponding OC/EC [i.e., $(OC/EC)^\#$] was calculated as
$(TC_{untreated} - EC_{extracted})/ EC_{extracted}$, where $TC_{untreated}$ indicates the total carbon concentration
measured before the extraction. As shown in Figure 6 for the wintertime HV samples, $(OC/EC)^*$
exhibited reasonable accordance with SNA/CO ($r = 0.72$) but $(OC/EC)^\#$ did not ($r = 0.02$). The clear
association between $(OC/EC)^*$ and SNA/CO, which was also supported by the results from LV ($r =$
0.66; Figure S3), provided additional evidence for the robustness of $EC_{untreated}$ determined by
IMPROVE-A.

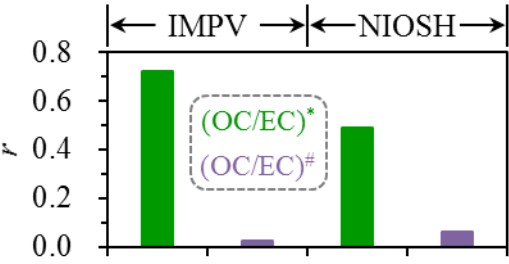

**Figure 6.** Comparison of $r$ values derived from the linear regressions of various OC/EC estimations on SNA/CO, based on the wintertime HV samples. IMPV indicates the IMPROVE-A temperature protocol. A total of four sets of OC/EC ratios were determined using different protocols and pre-treatment approaches. The OC/EC ratio measured by the untreated samples using IMPROVE-A, i.e., the IMPV-based (OC/EC)*, exhibited the strongest association with SNA/CO.

Compared to IMPROVE-A, NIOSH led to weaker correlations between (OC/EC)* and SNA/CO, as indicated by the smaller $r$ values determined (0.49 vs. 0.72 for HV and 0.18 vs. 0.66 for LV; Figures 4 and S3). In addition, the NIOSH-based (OC/EC)# did not exhibit apparent dependence on SNA/CO either ($r$ = 0.06 and 0.34 for HV and LV, respectively). Thus Figures 6 and S3 clearly reflected the limitations of NIOSH-based OC/EC and thus NIOSH-based EC, further highlighting the benefit of using $EC_{untreated}$ determined by IMPROVE-A.

**3.2 Evaluation of EC results from the spring campaign**

In this section we evaluated the EC results from April, also starting with the HV samples analyzed by IMPROVE-A. To highlight the role of agricultural fires, we first separated the April samples into two groups (namely the fire-impacted and typical samples), which were characterized by considerable and insignificant impacts of open burning, respectively. As described in Supporting Information, the criteria for fire-impacted samples could be simplified as a levoglucosan to $TC_{untreated}$ ratio ($f_{LG}$; on a basis of carbon mass) of above 1.8%, based on a synthesis of $f_{LG}$, the levoglucosan to water-soluble potassium ratio (LG/K$^+$) and satellite-based fire hotspots (Figure S4). Before filter extraction, the dependence of ATN on $EC_s$ could be approximated by a liner function (with a slope of 33.4 $\pm$ 1.5 m$^2$/gC and a close-to-zero intercept; $r$ = 0.97) for all the typical samples

and the majority of the fire-impacted ones (Figure 7a), leading to an $EC_{max}$ of 8 µgC/cm². For three
of the fire-impacted samples, $EC_s$ exceeded this threshold value and the ATN vs. $EC_s$ relationships
were found to deviate significantly from the regression line, especially for the two samples with $EC_s$
above 10 µgC/cm² (as highlighted in Figure 7a).

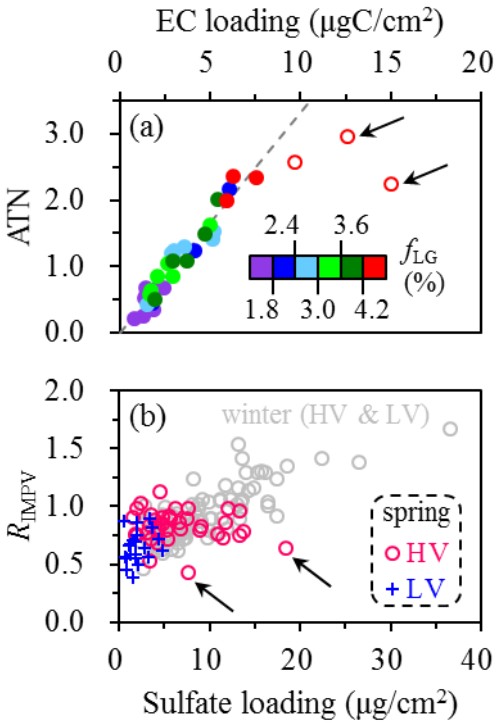


**Figure 7.** **(a)** Dependence of ATN on EC loading ($EC_s$) for the untreated HV samples collected in
spring, color-coded by $f_{LG}$ levels. Samples with linear and non-linear ATN vs. $EC_s$ dependence are
shown by the solid and open circles, respectively. $f_{LG}$ values higher than 1.8% indicated significant
impacts of agricultural fires. **(b)** Dependence of $R_{IMPV}$, i.e., the $EC_{extracted}$ to $EC_{untreated}$ ratio, on
sulfate loading in spring. Results from winter were also shown for comparison. In general, a
consistent relationship was observed between $R_{IMPV}$ and sulfate loading for the samples, including
both HV and LV, from different seasons. Two HV samples collected in spring were identified as
outliers, as highlighted by the arrows. The outlier samplers were also highlighted in (a),
corresponding to the two points showing significant non-linear dependences of ATN on EC loading.
All the EC results involved were measured by IMPROVE-A.
To elucidate factors responsible for the observed non-linear dependence of ATN on $EC_s$, we
compared EC results from the untreated and extracted filters. Given the relatively low sulfate
loadings observed throughout April (Figure 7b), it was with expectation that EC generally decreased
after the extraction. After excluding the results from two outliers (Figure 7b), the $EC_{extracted}$ to
$EC_{untreated}$ ratios averaged 0.84 ±0.11, comparable to results from the reference group in winter. The
two outlier samples, which were collected on the nights of April 10 and 20, 2021, showed $EC_{extracted}$
to $EC_{untreated}$ ratios of as low as 0.64 and 0.43, respectively. Such significant extraction-induced
decreases in EC could hardly be explained by the interference from sulfate in thermal-optical
analysis of the extracted filters (Figure 7b). Instead, the two outlier samples were found to show
several noteworthy features: (i) they corresponded to the two samples showing significant non-
linear ATN vs. $EC_s$ dependences before the extraction (Figure 7a); (ii) their $f_{LG}$ levels were at the
higher end of the fire-impacted samples (with levoglucosan concentrations exceeding 7 $\mu g/m^3$),
indicating extremely strong impacts of open burning (Figure 7a); (iii) their $LG/K^+$ ratios (> 1.7)
were also at the higher end of the fire-impacted samples, which were characteristic of the emissions
from smoldering combustion (Gao et al., 2003; Sullivan et al., 2019); (iv) their ATN decreased
apparently after the extraction, by ~1.0 which were about one order of magnitude higher than results
from the typical samples (~0.1). Thus it was inferred that $EC_{untreated}$ of the outlier samples likely
involved some light-absorbing organic compounds (i.e., BrC) emitted by agricultural fires with
relatively low combustion efficiencies, and the absorption capacities of these organics were strong
enough to make them a considerable contributor to ATN measured at 632 nm. Indeed, the BrC-
related overestimation of elemental carbon mass was expected to be reduced considerably after the
extraction. However, such overestimation seemed significant only for the two outlier samples
(Figure 7a). In addition, recalling the lower-than-one $EC_{extracted}$ to $EC_{untreated}$ ratios observed for the
other April samples (i.e., the sulfate-related artifact raised for the extracted filters), the methanol
extraction actually brought little benefit for the determination of EC by IMPROVE-A.
Unlike HV, all the LV samples showed a consistent relationship between ATN and $EC_s$ before
the extraction (Figure S5). It appeared that the strongly absorbing organics that could interfere EC
measurement were mainly concentrated in some agriculture-fire smoke emitted at night (as
indicated by the two outlier HV samples), whereas their influence on EC determination was
considerably weakened for the 24-h integrated LV samples. Thus the linear ATN vs. $EC_s$ dependence,
which was valid for all the untreated LV samples analyzed by IMPROVE-A, provided little evidence
for the necessity of methanol extraction.
We also investigated the OC/EC vs. $f_{LG}$ relationship for the HV samples collected in April.
Open burning was considered to be favorable for the increase of ambient OC/EC, since the aerosols
emitted were frequently found to be almost entirely organic (Liu et al., 2016; Garofalo et al., 2019;
Gkatzelis et al., 2024). Then it was not surprising to observe a moderate correlation between
$(OC/EC)^*$ and $f_{LG}$ ($r = 0.59$; Figure 8), i.e., an increasing trend of $(OC/EC)^*$ with stronger impacts
of agricultural fires. In addition, $(OC/EC)^*$ also depended moderately on SNA/CO ($r = 0.49$; Figure
8). This was with expectation as well, given the observational evidence on the concurrent
enhancements of secondary inorganic and organic aerosols (e.g., Liu et al., 2020; Cheng et al., 2022).
Replacing $(OC/EC)^*$ with $(OC/EC)^\#$ did not effectively strengthen the association of OC/EC with
$f_{LG}$ or SNA/CO (Figure 8). This conclusion also held for the LV samples (Figure 8). In summary,
we did not observe additional evidence supporting the incorporation of methanol extraction with
IMPROVE-A.

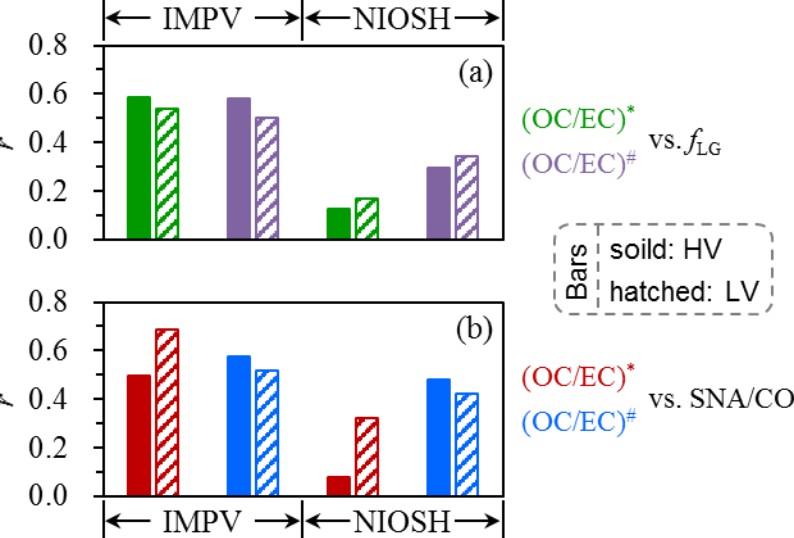

**Figure 8.** Comparisons of $r$ values derived from the linear regressions of various OC/EC estimations on **(a)** $f_{LG}$ and **(b)** SNA/CO, based on the spring samples. IMPV indicates the IMPROVE-A temperature protocol. Results from the HV and LV filters are shown by solid and hatched bars, respectively. For both HV and LV, a total of four sets of OC/EC ratios were determined using different protocols and pre-treatment approaches. In general, OC/EC ratios measured by the untreated samples using IMPROVE-A, i.e., the IMPV-based (OC/EC)*, exhibited reasonable associations with aerosol sources. Using other OC/EC estimations failed to or did not effectively enhance the associations.

Compared to (OC/EC)* determined by IMPROVE-A, the NIOSH-based (OC/EC)* and (OC/EC)# were less indicative of aerosol sources which could be reflected by $f_{LG}$ and SNA/CO. This was the case for both the HV and LV samples (Figure 8). Based on the discussions above, $EC_{untreated}$ determined by IMPROVE-A ($EC^*$) was also recommended for the conditions with prevalence of agricultural fires (i.e., April), in line with the conclusion derived for winter. In addition, it should be kept in mind that $EC^*$ could overestimate elemental carbon mass due to the interference from strongly absorbing BrC. However, such overestimation was generally uncommon, i.e., was significant only for some nighttime samples under extremely strong influences of low-efficiency agricultural fires (as indicated by the two HV samples identified as outliers; Figure 7).

**3.3 Comparison of measurement results from the HV and LV samplers**

As mentioned in the Methods section, each LV sample corresponded to a pair of daytime and

nighttime HV samples, indicating that measurement results from the HV samples could be averaged
and then compared to those determined by LV. Here the inter-sampler comparison was performed
for several components that are of broad interest in field observations, including elemental and
organic carbon, secondary inorganic ions (sulfate, nitrate and ammonium), other water-soluble ions
(potassium and chloride), organic tracer for biomass burning (levoglucosan), BrC mass
concentration and light absorption coefficient. Based on the evaluation results in previous sections,
$EC^*$ and the corresponding OC, i.e., $OC^*$ (measured before the extraction using IMPROVE-A), were
selected for the comparison. In addition, following Cheng et al. (2024), BrC mass was calculated as
the difference in TC between the untreated and extracted filters, while BrC absorption was
investigated at a wavelength of 365 nm, i.e., $(b_{abs})_{365}$.
As shown in Figure 9, the two samplers generally led to comparable measurement results for
all the species investigated. For example, the LV-to-HV ratios typically fell within the range of 0.8–
1.2, i.e., results from the two samples generally agreed within $\pm 20\%$. However, it was noticed that
$K^+$ was the only component with the majority of the LV results lower than HV. In addition, the LV-
to-HV ratio of $K^+$ was found to depend positively on the ratio of $PM_{2.5}$ to $PM_{10}$ (Figure 10a). The
$PM_{2.5}$ to $PM_{10}$ ratio was strongly associated with the influence of dust, typically exhibiting a
decreasing trend as the impact of dust became stronger (Putaud et al., 2010). Thus the events with
decreased LV-to-HV ratios of $K^+$ (all of which occurred in spring) presumably coincided with dust
episodes, when relatively large particles were expected to be a non-negligible contributor to $K^+$
(Wang et al., 2014a). Then a likely cause for the dependence shown in Figure 8a was that the
impactor performances (e.g., the size-cut curves) were not exactly the same for the two samplers,
such that some relatively large particles, if present, could be more effectively collected onto the HV
filters compared to LV.

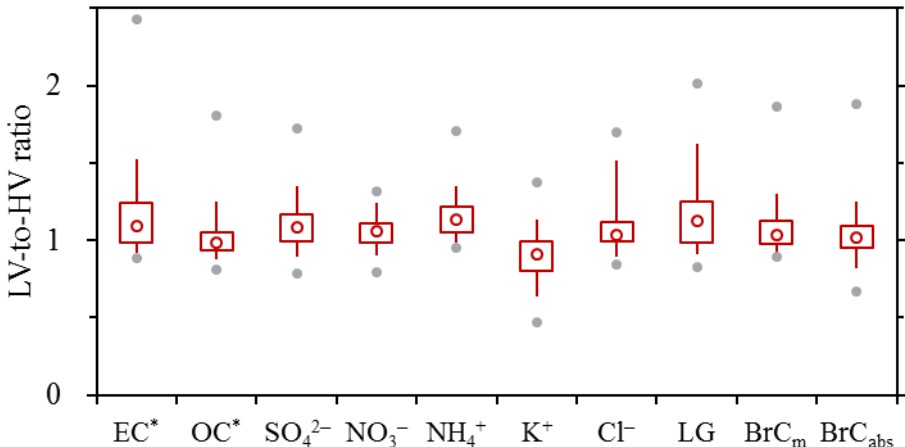

**Figure 9.** The LV-to-HV ratios determined for various species. $BrC_m$ and $BrC_{abs}$ indicate the mass concentration and $(b_{abs})_{365}$ of brown carbon, respectively.

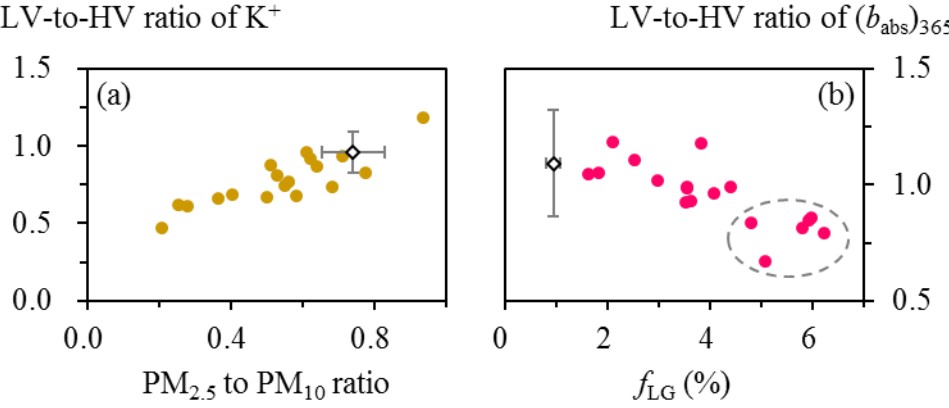

**Figure 10**. **(a)** Dependence of the LV-to-HV ratio of $K^+$ on the $PM_{2.5}$ to $PM_{10}$ ratio. **(b)** Dependence of the LV-to-HV ratio of $(b_{abs})_{365}$ on $f_{LG}$. Results from spring are shown on a sample-by-sample basis, as indicated by the solid circles. For winter, only the average results are shown as indicated by the diamonds. In (b), the events with substantially lower LV-to-HV ratios of $(b_{abs})_{365}$ are highlighted by the dashed oval.

In addition, it was noticed that although the two samplers led to generally comparable $(b_{abs})_{365}$ results, the corresponding LV-to-HV ratios decreased substantially (e.g., down to 0.67) for the episodes with extremely strong impacts of agricultural fires (Figure 10b). A possible explanation was that some fire-emitted chromophores were associated with relatively large particles that would be more effectively collected onto the HV filters. Although these chromophores represented an

important contributor to BrC absorption, their influence on BrC mass was likely insignificant, as
indicated by the little influence of agricultural fires on the LV-to-HV ratio of BrC mass (Figure S6).

Finally, it should be noted that for the species except $K^+$ and $(b_{abs})_{365}$, their LV-to-HV ratios

still showed different patterns of distribution (Figure 9). Analytical uncertainties should be partially
responsible, e.g., as indicated by the more significant variations in the LV-to-HV ratios of $EC^*$
compared to those of $OC^*$. Another likely cause was that the sampling duration of a given LV sample
(24 hours) actually did not equal that of the corresponding HV (15 hours), which would result in
different LV-to-HV ratios for the species with different diurnal cycles. In summary, many factors
could be responsible for the inter-sampler discrepancies shown in Figure 9. Typically, the overall
effects of these factors were higher LV-based concentrations than HV, with median LV-to-HV ratios
concentrating in a relatively narrow range of 1−1.15. However, this comparability may not always
hold. A possible explanation was that the impactor performances were more or less different
between the two samplers, thus for specific components such as $K^+$ and some fire-emitted
chromophores, this difference could exert a significant influence on their sampling.
**4. Conclusions and implications**

For the first time, EC results were compared among different sampling and analytical

approaches based on field observations in Northeast China. Two samplers with flow rates of 1.13
$m^3$/min and 5 L/min were operated together during two distinct seasons, whereas thermal-optical
analysis was performed by applying different protocols to both the untreated and extracted filters.
Results from different seasons jointly suggested that $EC_{extracted}$ measured by IMPROVE-A (i.e.,
IMPV) was biased by complex artifacts associated with sulfate. The IMPV-based $EC_{extracted}$ tended
to underestimate elemental carbon mass when sulfate was less abundant, whereas overestimations
were evident at sufficiently high loadings of sulfate. The turning point for the artifact shifting from
an underestimation to overestimation occurred in the sulfate loading range of 10–15 μg/cm$^2$. Such
high sulfate loadings were rarely encountered by the LV samples and thus the corresponding IMPV-
based EC typically decreased after the extraction (by ~20%). For the HV samples, their sulfate
covered a wide range of ~2–35 μg/cm$^2$ during winter and in this case, the extraction-induced
changes in EC usually varied between −50% and +50% for the IMPV-based results. In addition to
the complex sulfate-related artifacts, another problem identified for the IMPV-based EC$_{extracted}$ was
that the corresponding OC/EC ratio sometimes exhibited no association with the SNA/CO ratio,
which was used as an indicator for the significance of secondary aerosol production. The NIOSH-
based OC/EC ratios, determined by either EC$_{extracted}$ or EC$_{untreated}$, had the same problem. Then the
IMPV-based EC$_{untreated}$ (EC$^*$) was recommended, as the corresponding OC/EC could always be
reasonably linked to aerosol sources, e.g., secondary aerosol formation in winter and agricultural
fire impacts (as reflected by $f_{LG}$) in spring.
Inter-sampler comparisons were performed for various species that are of broad interest in field
observations, including EC$^*$. Although the flow rates differed by more than two orders of magnitude
between the two samplers, the LV and HV generally led to comparable measurement results for the
majority of the species, with the median LV-to-HV ratios falling within a relatively narrow range of
1–1.15.
This study also raised two points that merit attentions. The first is that the IMPV-based
EC$_{untreated}$, which was suggested for investigations on aerosol sources, might overestimate elemental
carbon mass under extremely strong impacts of open burning. This problem was attributed to
strongly absorbing BrC emitted by agricultural fires with relatively low combustion efficiencies.
Although uncommon for the ambient samples in Harbin (a megacity located in a main agricultural
region in China), this problem could introduce substantial uncertainties to the emission factors and
thus inventories of biomass burning. We suggested that a key step to refine the EC measurement
results was to concurrently minimize the inferences from strongly absorbing BrC and scattering
components. Methanol extraction followed by water extraction of filter samples was expected to be
a practical approach, which was worthy of further evaluations.
The second point was that some specific components should be interpreted with caution, even
when comparing their measurement results from samplers with the same nominal cut-point. Our
observational results indicated that some relatively large particles, if present, could be more
effectively collected by the high-volume $PM_{2.5}$ sampler compared to the low-volume one. This
problem was attributed to the fact that the inlet performances (e.g., the size-cut curve) could not be
exactly the same between the HV and LV $PM_{2.5}$ samplers. Among the various species involved in
this study, this problem affected the measurements of $K^+$ as well as some fire-emitted chromophores
in the fine mode. Thus we suggested that the $K^+$ results derived from different $PM_{2.5}$ samplers may
not be directly comparable. In addition, although we could not quantitatively determine the dust
contribution to $K^+$ measured by the high-volume $PM_{2.5}$ sampler, $K^+$ should be used with caution as
a biomass burning tracer for source apportionment studies relying on the HV-based observations.
By evaluating the observational results from different measurement approaches for species
commonly used in source apportionment, this study contributed to the understanding of aerosol
sources in Northeast China and thus the development of efficient haze pollution control strategies.
Using the recommended OC and EC results together with levoglucosan (rather than $K^+$) as the
biomass burning tracer, positive matrix factorization (PMF) analysis was performed for the 2020–
2021 heating season based on a total of ~200 LV samples (including those involved in this study;
Cheng et al., 2024). The concentrations of primary OC resolved were found to be in reasonable
agreement with those predicted by an air quality model, when agricultural fires were absent. The
consistency laid the foundation for the control policy focusing on primary aerosols. Cheng et al.
(2024) also found that the model failed to reproduce the observed SOA levels, with large
underestimations (by ~80%). Thus the observation-based source apportionment results were
currently irreplaceable for evaluating the benefits of reducing SOA precursors. In summary, this
study highlighted the importance of inter-method comparison for aerosol components (e.g., EC and
$K^+$) that are of broad interest in field observations. Such efforts are expected to be more urgently
needed for Northeast China, since this distinct region was recently targeted by the latest national-
level pollution control policy in China (State Council, 2021) and thus is facing stronger demand for
reducing $PM_{2.5}$.
**Data availability.** Data are available from the corresponding author upon request
(jiumengliu@hit.edu.cn).
**Author contributions.** YC and JL designed the study and prepared the paper, with inputs from all
the co-authors. YZ, ZZ and XC carried out the experiments.
**Competing interests.** The authors declare that they have no conflict of interest.
**Disclaimer.** Publisher's note: Copernicus Publications remains neutral with regard to jurisdictional
claims made in the text, published maps, institutional affiliations, or any other geographical
representation in this paper. While Copernicus Publications makes every effort to include
appropriate place names, the final responsibility lies with the authors.
**Acknowledgements.** The authors thank Zhen-yu Du at the National Research Center for
Environmental Analysis and Measurement and Lin-lin Liang at the Chinese Academy of
Meteorological Sciences for their help in sample analysis.
**Financial support.** This research has been supported by the National Natural Science Foundation
of China (42222706), the Natural Science Foundation of Heilongjiang Province (YQ2024D011),
the State Key Laboratory of Urban Water Resource and Environment (2023DX10) and the
Fundamental Research Funds for the Central Universities.

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
