# Peer review of "Technical note: Towards a stronger observational support for haze pollution control by"

_EGUsphere, 2025_

## Author Comment (AC2)

*General comments*

The manuscript by Cheng et al presents a detailed study in the measurement of carbonaceous aerosols by thermal-optical method. The authors put in a great deal of efforts in interpreting the experimental results. The analysis is generally sound and original. I support publication after my following comments are addressed.

*Major points*

**(1)** My main suggestion for the authors to consider is that it would be very helpful for the readers to better understand the discussion related to different scenarios by showing some thermograms of the paired samples, treated and untreated, with high/low sulfate. For example, it is not clear to me whether OC still presents in the extracted samples. Relevant information are scattered throughout the paper and therefore difficult to piece together.

**Our responses:** Thermograms were added as suggested. First, example thermograms (Figure R1) were provided in the "Methods" section to illustrate the influences of methanol extraction on OC and EC measurements.

[Figure]

**Figure R1.** Thermograms measured using a pair of untreated and extracted HV filters. The sample was collected during the daytime of 25 January, 2021. Temperature protocol used was NIOSH, in which the filter was heated in a He (first to 870 ℃ stepwise and then cooled down to 550 ℃) and a He/O$_2$ (from 550 to 890 ℃ stepwise) atmosphere sequentially. NIOSH had fixed durations for the various heating stages and thus was preferred for the comparison of thermograms. $I$ indicates the filter transmittance signal; FID indicates the carbon signal, which was measured by a flame ionization detector. The subscripts "untreated" and "extracted" distinguished the thermograms measured

before and after the extraction, while the split points of OC and EC were marked by the arrows. This figure was presented as Figure 1 in the revised manuscript.

Second, example thermograms (Figure R2) were provided in Section 3.1 to support our inference on the complex effects of sulfate on the measurement of $EC_{extracted}$.

[Figure]

**Figure R2.** Comparisons of (a) the evolution patterns of filter transmittance signal ($I$) and (b) the fractions of char-OC (i.e., $F_{char-OC}$) in He/O$_2$ carbon for two extracted HV samples with relatively high and low sulfate loadings (namely Sample-A and Sample-B, respectively). The two samples were collected during the daytime of 25 January and the nighttime of 6 January, 2021, respectively. They had sulfate loadings of 13.21 and 3.29 μg/cm$^2$, and organic carbon loadings of 3.10 and 0.86 μgC/cm$^2$, respectively. The temperature protocol used was IMPROVE-A, in which the filter was first heated to 580 ℃ in a He atmosphere and then to 840 ℃ in a He/O$_2$ atmosphere. The two modes had 4 (i.e., OC1 to OC4) and 3 (i.e., EC1 to EC3) heating stages, respectively. He/O$_2$ carbon indicated the amount of carbon evolving during the oxidizing mode, and was typically comprised of char-OC and EC for IMPROVE-A. This figure was presented as Figure 4 in the revised manuscript.

The related discussions were expanded accordingly: "*The influence was inferred to be dominated by backward scattering with relatively high sulfate loadings (e.g., for the targeted group), whereas by forward scattering when sulfate was less abundant (e.g., for the reference group). This inference was supported by the comparison of evolution patterns of filter transmittance signal under different sulfate loadings (Figure 4). For the extracted filter with abundant sulfate (i.e., Sample-A in Figure 4), the transmittance signal was largely unchanged during the He mode despite the sufficient organic carbon loading. Correspondingly, the operationally-defined char-OC only accounted for a*

*relatively small fraction of the carbon evolving during the He/O₂ mode (i.e., He/O₂ carbon). A possible explanation was that as the sample was heated, the drop of I induced by char-OC was compensated by the increase of I due to the reduction in sulfate-driven backward scattering. For the extracted filter with relatively small amount of sulfate (i.e., Sample-B in Figure 4), however, the transmittance signal decreased significantly during the He mode, and the char-OC contribution to He/O2 carbon became more considerable correspondingly. Given the much lower organic carbon loading for this sample (e.g., ~70% lower than Sample-A), the decrease of I was likely contributed by not only the formation of char-OC but also the reduction in sulfate-driven forward scattering. The EC$_{extracted}$ results appeared to be biased by different artifacts in the high- and low-sulfate cases, resulting in overestimations or underestimations of elemental carbon mass, respectively*".

**(2)** It would also be nice to include some discussion in the conclusion on how various results from this study can better inform the decision-making regarding haze pollution control strategy in the Northeastern China.

**Our responses:** Related discussions were added as suggested: "*By evaluating the observational results from different measurement approaches for species commonly used in source apportionment, this study contributed to the understanding of aerosol sources in Northeast China and thus the development of efficient haze pollution control strategies. Using the recommended OC and EC results together with levoglucosan (rather than K⁺) as the biomass burning tracer, positive matrix factorization (PMF) analysis was performed for the 2020–2021 heating season based on a total of ~200 LV samples (including those involved in this study; Cheng et al., 2024). The concentrations of primary OC resolved were found to be in reasonable agreement with those predicted by an air quality model, when agricultural fires were absent. The consistency laid the foundation for the control policy focusing on primary aerosols. Cheng et al. (2024) also found that the model failed to reproduce the observed SOA levels, with large underestimations (by ~80%). Thus the observation-based source apportionment results were currently irreplaceable for evaluating the benefits of reducing SOA precursors. In summary, this study highlighted the importance of inter-method comparison for aerosol components (e.g., EC and K⁺) that are of broad interest in field observations. Such efforts are expected to be more urgently needed for Northeast China, since this distinct region was recently targeted by the latest national-level pollution control policy in China (State Council, 2021) and thus is facing stronger demand for reducing PM$_{2.5}$*".

***Specific points***

**(1)** Line 24: it's not clear what the authors mean by "not directly comparable"; suggest brief clarification here.

**Our responses:** As suggested by Referee #1, this sentence was re-written as: "*We suggested that $K^+$ and light absorption coefficients of brown carbon should be compared or integrated with caution across studies using different $PM_{2.5}$ samplers*".

**(2)** Line 47: giving rise to -> producing

**Our responses:** The change was made as suggested.

**(3)** Line 61: suggest to elaborate on the different techniques for EC measurements

**Our responses:** The EC measurement techniques were clarified as suggested: "*since different analytical methods (i.e., NIOSH vs. IMPROVE-A temperature protocols) for EC were deployed*".

**(4)** Line 70: [EC] survives to -> I think you are trying to say that EC "evolves from the filter at [a higher temperature than OC]?

**Our responses:** This sentence was changed to "*EC evolves form the filter at higher temperatures than OC*" to make it easier to follow.

**(5)** Line 113: 00:00-00:00 sampling for low volume filters?

**Our responses:** The sampling time was clarified for the LV sampler: "*the LV one was operated on a daily basis (~09:00–09:00 of the next day), leading to 24-h integrated samples*".

**(6)** Line 116: do you mean the high particle loading would saturate the detection of OC/EC analyzer? What is the upper detection limit of the analyzer?

**Our responses:** This sentence was re-written as "*One reason for the relatively short sampling durations of HV was to avoid high particle loadings that could prohibit proper filter transmittance measurement*" to make it more clear. The nominal upper detection limit was 750 $\mu gC/cm^2$. In reality, however, the filter transmittance detector could be saturated at substantially lower carbon loadings, since the filter transmittance signal was also influenced by the inorganic scattering components such as sulfate and nitrate.

**(7)** Figure 1. How is TC for the methanol extracted samples in Figure 1 calculated?

Also are the scatters in the lower end of TC mostly from the extracted samples? What is the uncertainty introduced by extraction?

**Our responses:** For the extracted samples, TC was determined as the amount of carbon remained on the filter after extraction. In addition, the scatters in the lower end of TC indeed corresponded to the extracted samples, and the extraction-induced uncertainties were clearly stated in the revised manuscript: "*It should be noted that for the parallel TC and ATN measurements by different protocols, the relative standard deviation (RSD) levels indeed increased after the extraction, e.g., from ~2 to 5% and from ~2 to 4% for the HV samples, respectively. However, the RSD levels, i.e., the uncertainties, were considered low enough for both the untreated and extracted filters*".

**(8)** Figure 2. Blue circles in 2a are not properly labeled.

**Our responses:** The description of blue circles was added in the figure caption.

**(9)** Line 215-217: Have you done any experiment with water-extraction, since sulfate should be soluble in water and removed from the filters?

**Our responses:** Water-extraction was not performed here. The effects of step-wise extraction (methanol followed by water) on EC measurement will be investigated in our future studies.

**(10)** Line 414: more or less distributed on -> associated with

**Our responses:** The change was made as suggested.

---

## Author Response (AR1)

**Dear Editor,**

**Manuscript number:** egusphere-2025-537

**Title:** Technical note: Towards a stronger observational support for haze pollution control by interpreting carbonaceous aerosol results derived from different measurement approaches

Many thanks to you and the referees for the valuable comments and suggestions. We have considered the points raised and revised our manuscript accordingly. Our detailed responses and relevant changes are presented below.

*Comments from Reviewer #1*

*General comments*

This manuscript compared carbonaceous aerosol data measured by different sampling and analytical approaches, based on field campaigns (winter and spring) performed at an urban site in Northeast China. Major contributions included that the effects of methanol extraction on EC measurement were clearly explained, and the EC and OC results that were most indicative of aerosol sources, including secondary aerosol formation and open burning emissions, were identified. The authors also compared aerosol compositions measured by different PM2.5 samplers, and argued that the results were not always comparable. The methodologies and interpretations were generally reliable, and the conclusions provided implications for future studies on aerosol observation and source apportionment. I agree that the manuscript is suitable to be submitted as a technical note, rather than a research article. It should be publishable if the authors could properly address the following concerns. The comments were raised during the "quick reports" stage, and are now posted for open discussion.

*Specific points*

**(1)** Line 23-25. Tone down the statement. In the abstract, it is better to say that $K^+$ and brown carbon results should be compared or integrated with caution across studies using different $PM_{2.5}$ samplers.

**Our responses:** The sentence was re-written as: "*We suggested that $K^+$ and light absorption coefficients of brown carbon should be compared or integrated with caution across studies using different $PM_{2.5}$ samplers*" **(see lines 23-27)**.

**(2)** Lines 40-42. Clarify that the statements are for chemical transport models.

**Our responses:** We prefer to keep the sentence as is, since the term "model" used here is not limited to chemical transport models, e.g., it also includes dispersion models.

**(3)** Line 96. Re-write it as: …efforts on the exploration of $PM_{2.5}$ sources…

**Our responses:** The change was made as suggested **(see line 99)**.

**(4)** Line 101. Is the abbreviation necessary?

**Our responses:** The abbreviation "HIT" was removed **(see line 104)**.

**(5)** Line 108-111. The sentence is too long.

**Our responses:** The sentence was divided into separate ones as suggested: "*The 2021 winter campaign covered the entirety of January, the coldest month during that year with an average temperature of –19 °C. In addition, the spring campaign was conducted during 10–30 April of 2021...*" **(see lines 111-114)**.

**(6)** Section 2.2. To my understanding, the reason for using transmittance correction is to link EC and ATN. This point should be clearly explained.

**Our responses:** This point was clarified as suggested: "*This correction approach was applied since the intensity of the filter transmittance signal (I) has a clear association with EC, e.g., as assumed by the Aethalometer, another widely used instrument for measuring black carbon*" **(see lines 128-130)**.

**(7)** Line 130. Add "typically" before "with a linear dependence".

**Our responses:** The change was made as suggested **(see line 136)**.

**(8)** Line 191. Termed as?

**Our responses:** The mistake was corrected **(see line 212)**.

**(9)** Figure 2. Define "ref." appeared in the figure.

**Our responses:** We defined "ref." as suggested **(see line 213)**.

**(10)** Line 195. I guess "Group-R" was missing.

**Our responses:** The mistake was corrected **(see line 216)**.

**(11)** Line 213. Changing "these samples" to "the targeted samples" would make the

sentence clearer.

**Our responses:** The change was made as suggested **(see line 234)**.

**(12)** Line 216. Add "typically considered" before "soluble".

**Our responses:** The sentence was changed to: "*Other scattering components such as nitrate and secondary organic aerosol (SOA) were not discussed here, since they were typically considered soluble in methanol and should be absent in the extracted filters*" **(see lines 236-238)**.

**(13)** Line 264. Remove "distinct".

**Our responses:** The change was made as suggested **(see line 311)**.

**(14)** As can be seen from Figure 5b, after excluding the two highlighted samples shown by the arrows, the overestimation of BC mass by EC-untreated seemed more or less evident for the other HV samples with high sulfate loadings. However, I agree that such overestimations could be considered insignificant, since they did not disturb the linear dependence of ATN on EC-untreated shown in Figure 5a. Thus, the following changes are required: Figure 5 caption and the main text, define the two distinct samples as outliers; Line 337, change "apparent" to "significant"; Line 376, change "considerable" to "significant".

**Our responses:** The changes were made as suggested **(see lines 359-362, 367, 369, 372, 380, 385-386, 393, and 425-426)**.

**(15)** Lines 396-398. Confirm whether all the events occurred in the spring. If some of them occurred in winter, I am afraid that it is not robust enough to attribute the events to dust.

**Our responses:** We confirmed that all the events were encountered in the spring. This point was clarified in the revised manuscript **(see line 447)**.

**(16)** Lines 400-401. The statement needs to be refined. Based on the available results, it is more proper to say the impact performances were not exactly the same for the two samplers, and some large particles were more effectively collected by one of them. Lines 413-415 and the conclusions have the same problem.

**Our responses:** The two sentences were re-written as: "*the impactor performances (e.g., the size-cut curves) were not exactly the same for the two samplers, such that*

*some relatively large particles, if present, could be more effectively collected onto the HV filters compared to LV*" and "*some fire-emitted chromophores were associated with relatively large particles that would be more effectively collected onto the HV filters*". The conclusion was updated accordingly: "*some relatively large particles, if present, could be more effectively collected by the high-volume PM$_{2.5}$ sampler compared to the low-volume one*" **(see lines 450-452, 465-467, and 525-527)**.

**(17)** Lines 444-449. The statements were difficult to follow. Re-organize them.

**Our responses:** The sentences were re-written as: "*In addition to the complex sulfate-related artifacts, another problem identified for the IMPV-based EC$_{extracted}$ was that the corresponding OC/EC ratio sometimes exhibited no association with the SNA/CO ratio, which was used as an indicator for the significance of secondary aerosol production. The NIOSH-based OC/EC ratios, determined by either EC$_{extracted}$ or EC$_{untreated}$, had the same problem*" **(see lines 495-504)**.

*Comments from Reviewer #2*

*General comments*

The manuscript by Cheng et al presents a detailed study in the measurement of carbonaceous aerosols by thermal-optical method. The authors put in a great deal of efforts in interpreting the experimental results. The analysis is generally sound and original. I support publication after my following comments are addressed.

*Major points*

**(1)** My main suggestion for the authors to consider is that it would be very helpful for the readers to better understand the discussion related to different scenarios by showing some thermograms of the paired samples, treated and untreated, with high/low sulfate. For example, it is not clear to me whether OC still presents in the extracted samples. Relevant information are scattered throughout the paper and therefore difficult to piece together.

**Our responses:** Thermograms were added as suggested. First, example thermograms (Figure R1) were provided in the "Methods" section to illustrate the influences of methanol extraction on OC and EC measurements **(see lines 148-157)**.

[revised manuscript text omitted]

*Specific points*

**(1)** Line 24: it's not clear what the authors mean by "not directly comparable"; suggest brief clarification here.

**Our responses:** As suggested by Referee #1, this sentence was re-written as: "*We suggested that K$^+$ and light absorption coefficients of brown carbon should be compared or integrated with caution across studies using different PM$_{2.5}$ samplers*" **(see lines 23-27)**.

**(2)** Line 47: giving rise to -> producing

**Our responses:** The change was made as suggested **(see line 49)**.

**(3)** Line 61: suggest to elaborate on the different techniques for EC measurements

**Our responses:** The EC measurement techniques were clarified as suggested: "*since different analytical methods (i.e., NIOSH vs. IMPROVE-A temperature protocols) for EC were deployed*" **(see lines 63-64)**.

**(4)** Line 70: [EC] survives to -> I think you are trying to say that EC "evolves from the filter at [a higher temperature than OC]?

**Our responses:** This sentence was changed to "*EC evolves form the filter at higher temperatures than OC*" to make it easier to follow **(see lines 73-74)**.

**(5)** Line 113: 00:00-00:00 sampling for low volume filters?

**Our responses:** The sampling time was clarified for the LV sampler: "*the LV one was operated on a daily basis (~09:00–09:00 of the next day), leading to 24-h integrated samples*" **(see line 116)**.

**(6)** Line 116: do you mean the high particle loading would saturate the detection of OC/EC analyzer? What is the upper detection limit of the analyzer?

**Our responses:** This sentence was re-written as "*One reason for the relatively short sampling durations of HV was to avoid high particle loadings that could prohibit proper filter transmittance measurement*" to make it more clear **(see lines 118-120)**. The

nominal upper detection limit was 750 μgC/cm². In reality, however, the filter transmittance detector could be saturated at substantially lower carbon loadings, since the filter transmittance signal was also influenced by the inorganic scattering components such as sulfate and nitrate.

**(7)** Figure 1. How is TC for the methanol extracted samples in Figure 1 calculated? Also are the scatters in the lower end of TC mostly from the extracted samples? What is the uncertainty introduced by extraction?

**Our responses:** For the extracted samples, TC was determined as the amount of carbon remained on the filter after extraction. In addition, the scatters in the lower end of TC indeed corresponded to the extracted samples, and the extraction-induced uncertainties were clearly stated in the revised manuscript: "*It should be noted that for the parallel TC and ATN measurements by different protocols, the relative standard deviation (RSD) levels indeed increased after the extraction, e.g., from ~2 to 5% and from ~2 to 4% for the HV samples, respectively. However, the RSD levels, i.e., the uncertainties, were considered low enough for both the untreated and extracted filters*" **(see lines 137-141)**.

**(8)** Figure 2. Blue circles in 2a are not properly labeled.

**Our responses:** The description of blue circles was added in the figure caption **(see lines 210-211)**.

**(9)** Line 215-217: Have you done any experiment with water-extraction, since sulfate should be soluble in water and removed from the filters?

**Our responses:** Water-extraction was not performed here. The effects of step-wise extraction (methanol followed by water) on EC measurement will be investigated in our future studies.

**(10)** Line 414: more or less distributed on -> associated with

**Our responses:** The change was made as suggested **(see line 465)**.

Again, we thank the referees very much for their valuable comments and suggestions.

Sincerely yours,
Jiu-meng Liu, PhD (jiumengliu@hit.edu.cn)
School of Environment, Harbin Institute of Technology